# Hospital length of stay throughout bed pathways and factors affecting this time: A non-concurrent cohort study of Colombia COVID-19 patients and an unCoVer network project

Lina Marcela Ruiz Galvis[1,2☯]*, Carlos Andrés Pérez Aguirre[2,3☯], Juan Pablo Pérez Bedoya[2‡], Oscar Ignacio Mendoza Cardozo[2‡], Noël Christopher Barengo[4,5‡], Juan Pablo Sánchez Escudero[2‡], Johnatan Cardona Jiménez[3‡], Paula Andrea Diaz Valencia[2‡]

1 Grupo de Fundamentos y Enseñanza de la Física y los Sistemas Dinámicos, Natural Science Department, Universidad de Antioquia, Medellín, Colombia, 2 Epidemiology Group at National College of Public Health, Universidad de Antioquia, Medellín, Colombia, 3 Escuela de Estadística, Universidad Nacional de Colombia, Medellín, Colombia, 4 Herbert Wertheim College of Medicine & Robert Stempel College of Public Health and Social Work, Florida International University, Miami, Florida, United States of America, 5 Faculty of Medicine, Riga Stradins University, Riga, Latvia

☯ These authors contributed equally to this work.
‡ JPPB, OIMC, NCB, JPSE, JCJ and PADV also contributed equally to this work.
* lina.ruiz2@udea.edu.co

**Data Availability Statement:** All the code for statistical analysis and databases used in this study

## Abstract

Predictions of hospital beds occupancy depends on hospital admission rates and the length of stay (LoS) according to bed type (general ward -GW- and intensive care unit -ICU- beds). The objective of this study was to describe the LoS of COVID-19 hospital patients in Colombia during 2020–2021. Accelerated failure time models were used to estimate the LoS distribution according to each bed type and throughout each bed pathway. Acceleration factors and 95% confidence intervals were calculated to measure the effect on LoS of the outcome, sex, age, admission period during the epidemic (i.e., epidemic waves, peaks or valleys, and before/after vaccination period), and patients geographic origin. Most of the admitted COVID-19 patients occupied just a GW bed. Recovered patients spent more time in the GW and ICU beds than deceased patients. Men had longer LoS than women. In general, the LoS increased with age. Finally, the LoS varied along epidemic waves. It was lower in epidemic valleys than peaks, and decreased after vaccinations began in Colombia. Our study highlights the necessity of analyzing local data on hospital admission rates and LoS to design strategies to prioritize hospital beds resources during the current and future pandemics.

## Introduction

SARS-CoV-2 is difficult to be eradicated, because of its ubiquity [1], the incomplete vaccine coverage [2, 3], and virus evolution [4]. Hence, ongoing strategies to deal with the endemic

can be accessed at https://github.com/
LinaMRuizG/unCoVer_ColombiaLoSCOVID-19. The
processing of raw data to get the databases used
are in https://github.com/LinaMRuizG/unCoVer_
processingData. All authors had full access to the
study data and accepted the responsibility to
submit it for publication.

**Funding:** This project was funded by the European
Union's Horizon 2020 Research and Innovation
Program (Grant Agreement No 101016216). The
funders do not played any roles in the study
design, data collection and analysis, decision to
publish, and preparation of the manuscript.

**Competing interests:** The authors have declared
that no competing interests exist.

presence of SARS-CoV-2 in populations over the long term will be needed [3]. It is important
to understand the impact of COVID-19 on hospital capacity and to be prepared for hospital
bed demand by COVID-19 patients [5]. High demand for beds could severely decrease the
quality of services in part because of the limited number of general ward (GW) and Intensive
Care Units (ICU) [6] beds. As a consequence, designing strategies to prioritize hospital beds
resources is mandatory to cope with new waves of the epidemic or endemic outbreaks [7].

One of the main indicators needed for any predictions is the estimation of the number of
hospital beds occupancy, which is essential in managing the resources allocation [8]. This indi-
cator is a function of hospital admissions rates and the total LoS in particular bed types (GW
and ICU beds) [7, 8]. It is possible to model the rate of hospitalization in many settings based
on estimated epidemic curves [7]. On the other hand, statistical models allow us to estimate
the LoS through the observation of patients' bed pathways (BPs), which is the sequence of
transfers between bed types during a hospital stay [6, 9]. Additionally, different factors such as
sex, age, comorbidities, and the variations in the treatments, medical staff, equipment, and
access to health services could potentially explain differences in the hospital LoS and the level
of care needed [9, 10]. However, up-to-date, there is little scientific information available on
the characteristics of hospital LoS of COVID-19 patients in Colombia. Therefore, the objective
of this study was to describe the hospital LoS of Colombia COVID-19 patients during 2020–
2021.

## Methodology

### Database description

Publicly available data of all registered COVID-19 hospital patients from March 15[th] 2020 to
August 17[th] 2021 were retrieved from the Colombia National Health Institute [11]. A person
was recorded in this database once it was diagnosed with COVID-19. This means, the track of
the patients' bed pathway is made from the moment they became positive for SARS-CoV-2.
The diagnosis of SARS-CoV-2 was confirmed in all patients by diagnostic tests [12]. From this
database, we only included in-hospital patients diagnosed with COVID-19 who either recov-
ered or deceased. Recovered patients were those with a negative test result and/or with 21 days
without symptoms since the symptoms onset [12]. This study was developed within the
unCoVer Horizon 2020 project framework [13].

This study followed the Good Clinical Practice guidelines and the guidelines of the Helsinki
Declaration. This study was a secondary data analysis using de-identified data downloaded
from the Colombian Institute of Public Health that is publicly available [11]. Ethical approval
was waived by IRB of the Faculty of Public Health, Universidad de Antioquia since the analysis
was considered nonhuman subject research.

### LoS by bed pathway and stage

From the database, COVID-19 patients in Colombia were classified as being in hospital or
ICU patients. It means, they could occupy two types of bed: a GW bed and ICU bed, respec-
tively. Unlike a GW bed, an ICU bed implies intensive and specialized medical and nursing
care, an enhanced capacity for monitoring, and multiple modalities of physiologic organ sup-
port [14]. Daily hospital bed locations by patient was retrieved and ordered chronologically to
identify all possible stages (i.e., bed locations) and BPs. For each patient the time of stay in
each stage of its BP was estimated [6]. The LoS of the hospitalization until the first discharge
was included in the analysis. Taking into account the maximum times previously reported in
the literature [6, 8–10], a right truncation of the data was made to restrict the maximum length
of hospital stay to 100 days.

An Accelerated Failure Time (AFT) model was run for each stage of every BP with different distributions (i.e., Lognormal, Gamma, and Weibull) [15, 16]. AFT model was used because the aim was to estimate the times to failure and the effect on time of some covariates. This is a parametric model in the statistical field of survival analysis that offers an alternative to the often used Proportional Hazard models [16]. For this survival analysis, the variable of interest was the amount of time measured in days that a patient occupies a type of bed. The Akaike Information Criterion was used to select the best fitting parametric distribution. For this survival analysis we used RStudio 2021.09.0.

We used the estimated parameters of each best fitting distribution for each stage in each BP to estimate the median of the LoS distribution and its 95% confidence interval [6]. For this, we implemented a Monte Carlo method (S1 Fig).

### Association between LoS and covariates

We run AFT models adjusted for all covariates for each one of the two most common BPs. We also run an AFT models adjusted for each covariate at time for the four most common BPs, which means that, for each covariate an AFT model was run for each stage of each bed pathway.

The AFT model estimates the accelerated factor ($\theta$) which allows to evaluate the effect of predictor variables on survival time just as the hazard ratio allows the evaluation of predictor variables on the hazard [16]. The covariates were BP, outcome (i.e., recovery or death), age (i.e., 1–25, 26–50, 51–75, and >75 years), sex (men *vs* women), and the epidemic phase in which the patient was admitted and discharge/death as explained below.

We identified some phases of the COVID-19 epidemic curve as explained in the SI. These phases are characterized in epidemiology as waves, and peaks or valleys as is detailed in S2 and S3 Figs, respectively. Then, each patient was classified according to a specific wave, and peak or valley depending on the dates in which they were admitted and discharged from the hospital or died. The study population was also categorized into patients admitted before and after the vaccination period in Colombia, which started in February 2021 [17].

Finally, an AFT model for the two most common BPs was calculated to estimate the association between the LoS and the geographic region of origin of each patient. The median of LoS for the best fitting distribution was estimated as previously described. Then, a k-means clustering algorithm was applied to group the geographical regions with similar medians. We also divided the distribution of the median LoS of geographical regions into quartiles and labelled the regions accordingly.

All the code for statistical analyses and databases used in this study can be accessed at GitHub. The processing of raw data to get the databases used are in GitHub. All authors had full access to the study data and accepted the responsibility to submit for publication.

## Results

### LoS by bed type and bed pathway

In total, 245,371 COVID-19 patients were hospitalised, they were recovered or deceased between March 15th 2020 and August 17th 2021 in Colombia. They represented 5.03% of the cumulative COVID-19 cases in Colombia over that same period. After identifying their BP and filtering those with LoS lower than 100 days, 228,426 COVID-19 patients remained to be included for the data analysis (Fig 1). The COVID-19 patients in Colombia could occupy two types of bed: a GW or ICU bed. Six different BPs were possible. We identified that the BPs have between one to three stages which represent the different bed transfers of a patient during its hospital stay, and noticed that BP1 or BP2 only have one stage (stage 1), but for the BP5 or BP6 the patients have three stages (stage 1 to 3) (Fig 1). Each one of the LoS distributions were

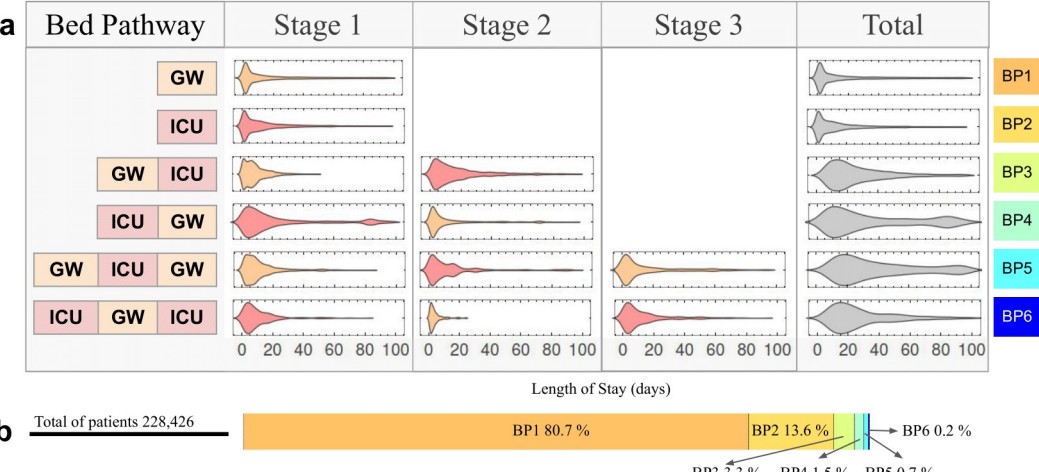

**Fig 1. Bed pathways of Colombia COVID-19 patients.** (**a**) For each bed pathway (GW—[BP1]; ICU—[BP2]; GW, ICU—[BP3]; ICU, GW—[BP4]; GW, ICU, GW—[BP5]; ICU, GW, ICU–[BP6]). It shows the LoS distribution for each stage/bed type (GW in light orange, ICU in pink) and the total of bed pathway (in gray). (**b**) The proportions of hospital admissions entering each bed pathway.

best fitted to the Lognormal distribution except the stage 1 and 2 of BP3 and the total distribution of BP4, these were best fitted to Gamma distribution.

Table 1 presents the median and IQR of the length of stay in each stage of each bed pathway. Most of the patients only occupied a GW bed (80.70%) or an ICU bed (13.58%), with median times of 6.5 (95% CI 6.2–6.7) and 6.7 (95% CI 6.5–6.9) days, respectively. The remaining patients occupied both hospital and ICU beds at different moments (5.7%).

## Association between LoS and some covariates

Table 2 shows that recovered patients spent 2.00 ($\theta$, 95% C.I 1.97–2.03) and 1.79 ($\theta$, 95% C.I 1.74–1.85) more time in GW (BP1) and ICU (BP2) beds, respectively, than deceased patients.

**Table 1. Median and IQR of the length of stay in each stage of each bed pathway.** These were estimated with the time distributions that best fitted according to the AFT models.

| Bed Pathway | n | % | Stage 1 | | Stage 2 | | Stage 3 | | Total | |
|---|---|---|---|---|---|---|---|---|---|---|
| | | | x | IQR | x | IQR | x | IQR | x | IQR |
| GW (BP1) | 184,340 | 80.7 | 6.48 (6.23–6.75) | 2.16–19.45 | NA | NA | NA | NA | 6.48 (6.23–6.75) | 2.16–19.45 |
| ICU (BP2) | 31,032 | 13.58 | 6.69 (6.46–6.94) | 2.56–17.53 | NA | NA | NA | NA | 6.69 (6.46–6.94) | 2.56–17.53 |
| GW,ICU (BP3) | 7,632 | 3.34 | 7.24 (7.07–7.43) | 3.44–13.27 | 12.38 (12.04–12.72) | 5.23–24.45 | NA | NA | 20.57 (20.17–20.97) | 12.16–34.77 |
| ICU,GW (BP4) | 3,341 | 1.46 | 10.68 (10.34–11.03) | 4.41–25.90 | 4.92 (4.75–5.08) | 1.95–12.46 | NA | NA | 26.98 (26.36–27.62) | 13.19–48.40 |
| GW,ICU,GW (BP5) | 1,653 | 0.72 | 6.77 (6.59–6.95) | 3.30–13.86 | 6.73 (6.53–6.94) | 2.86–15.82 | 5.79 (5.60–5.99) | 2.30–14.58 | 30.54 (29.99–31.08) | 18.68–49.90 |
| ICU,GW,ICU (BP6) | 428 | 0.19 | 6.47 (6.30–6.64) | 3.12–13.39 | 2.68 (2.62–2.74) | 1.45–4.92 | 6.37 (6.18–6.57) | 2.76–14.70 | 21.82 (21.44–22.21) | 13.38–35.59 |
| Total | 228,426 | 100 | | | | | | | | |

GW: General Ward, ICU: Intensive Care Unit, n: number of hospital admissions entering each bed pathway, %: proportion of hospital admissions entering each bed pathway, x: Median and 95% C.I., IQR: Interquartile range, NA: Not applicable.

**Table 2. Acceleration factors of the length of stay in each bed type and each bed pathway for a model adjusted by all covariates.**

| Covariate | GW (BP1) | ICU (BP2) |
|---|---|---|
| **Outcome** | 2.00 | 1.79 |
| **Recovery** | (1.97–2.03) | (1.74–1.85) |
| **Age** | 1.07 | 1.24 |
| **26–50** | (1.05–1.09) | (1.17–1.31) |
| | 1.26 | 1.44 |
| **51–75** | (1.24–1.28) | (1.36–1.52) |
| | 1.18 | 1.27 |
| **>75** | (1.16–1.21) | (1.19–1.36) |
| **Sex** | 1.04 | 1.07 |
| **Men** | (1.03–1.05) | (1.05–1.10) |
| **Waves** | 0.27 | 0.37 |
| **W2** | (0.26–0.28) | (0.35–0.39) |
| | 0.85 | 1.66 |
| **W3** | (0.83–0.87) | (1.58–1.75) |
| | 1.14 | 2.17 |
| **W4** | (1.07–1.21) | (1.89–2.49) |
| | 1.56 | 3.10 |
| **W5** | (1.47–1.67) | (2.68–3.60) |
| **Valleys/Peaks** | 0.58 | 0.68 |
| **Valley** | (0.57–0.59) | (0.66–0.70) |
| **Vaccination Period** | 0.57 | 0.59 |
| **Yes** | (0.53–0.61) | (0.52–0.68) |

GW: General Ward, ICU: Intensive Care Unit bed. The basal groups are Death (for outcome), <26 (for age), Women (for sex), W1 (for waves), Peaks (for Valleys and Peaks), No (for Vaccination period).

The LoS in both beds increased with age, but that increase was non-monotonic. Men spent more time than women in both beds (GW: 1.04 $\theta$, 95% C.I 1.03–1.05 and ICU: 1.07 $\theta$, 95% C.I 1.05–1.10). The LoS increased throughout epidemic waves but decreased in the vaccination period with respect to non-vaccination period. Finally, they were lower in valleys than peaks (GW: 0.58 $\theta$, 95% C.I 0.57–0.59 and ICU: 0.68 $\theta$, 95% C.I 0.66–0.70).

S1 Table presents the acceleration factors of the LoS for each stage of the four most common BPs, and S2 Table presents the times or LoS for each stage of the two most common BPs. The LoS of each bed type depended on the BP. For instance, patients in BP3 and BP4 spent 0.95 ($\theta$, 95% C.I 0.91–0.98) and 0.76 ($\theta$, 95% C.I 0.72–0.80) times less time in GW bed than those in BP1 (S1 Table). They have a GW bed LoS of 6.16 ($\theta$, 95% C.I 5.92–6.41) and 4.92 ($\theta$, 95% C.I 4.73–5.12) days, respectively (S2 Table). In contrast, patients in BP3 and BP4 spent 1.51 ($\theta$, 95% C.I 1.46–1.56) and 1.60 ($\theta$, 95% C.I 1.52–1.68) times more time in ICU beds than those in BP2.

The results for BP1 and BP2 with the model adjusted by all covariates and adjusted by one covariate at time were similar in the behavior of covariates except for the waves and the vaccination period. For instance, the LoS in each bed type and BP was significantly higher for recovered patients than for deceased ones, except for the GW bed (stage 2) in BP4 (S1 Table). Age had a significant effect in the LoS mainly for BP1 and BP2 (S1 Table). In general for both pathways, the LoS in the GW and ICU increased with age, but that increase was non-monotonic (S1 Table). Moreover, sex was associated with the LoS only for BP1 and BP2. Men had 1.11 ($\theta$,

95% C.I 1.09–1.98) and 1.22 (95% C.I 1.18–1.26) times the LoS in GW and ICU beds compared with women (S1 Table).

Additionally, the LoS were significantly different throughout the epidemic COVID-19 waves compared with the first wave in almost all BPs (S1 Table). The LoS in GW for the BP1 was lower in W2, W3, W4 and W5 than in W1 (i.e., $\theta$ 0.18, 95% C.I 0.17–0.18; $\theta$ 0.62, 95% C.I 0.60–0.63; $\theta$ 0.46, 95% C.I 0.45–0.47; and $\theta$ 0.81, 95% C.I 0.79–0.83, respectively), with the maximal diminution for W2. In contrast, the LoS in ICU for the BP2 was higher in W3 to W5 with respect to W1, only with a diminution of 0.35 ($\theta$, 95% C.I 0.33–0.37) in W2. The LoS for BPs was lower in valleys than peaks.

Contrary to the model adjusted by all covariates, for a model just adjusted by vaccination period we found that: for BP1 and BP2, the LoS was larger in the vaccination period than before it ($\theta$ 1.80, 95% C.I 1.77–1.84; $\theta$, 2.16, 95% C.I 2.09–2.23, respectively). Thus, the majority of patients occupying a GW or ICU bed spent more time in these beds after vaccination in the population began than before.

### Association between LoS and geographical region of origin

Fig 2A and 2B present the cluster analysis for patients' geographic region of origin according to the median of the LoS in GW and ICU bed for BP1 and BP2. There were five and four clustering for GW and ICU beds, respectively. Fig 2C and 2D present the regions classified by the quartiles of the median distribution of LoS. They clearly show the regions with both high GW and ICU LoS (i.e., Nariño, Putumayo, Chocó, and Casanare), and both low GW and ICU LoS (i.e., Meta, Guaviare, Vaupés, Guainía, Arauca). Also, people from certain regions had median LoS in the lowest/highest quartile in GW beds but the highest/lowest quartile in ICU beds (i.e., Sucre).

### Discussion

The Colombia COVID-19 patients occupied mainly a GW or ICU bed during the hospitalization between March 2020 to August 2021. The length of stay in each one of these beds depends on the outcome (i.e., recovery or death), age, sex, the epidemic period in which the patient gets admitted and discharge/death (i.e., epidemic waves, peaks/valleys, and before/after the vaccination period). There were also differences in LoS throughout the geographical regions from which the patients origin, which prompts further investigation into how the covariates mentioned previously are distributed geographically. For instance, recovered patients spent more time in the GW and ICU than deceased patients. Men had larger LoS for both GW and ICU beds than women. In general, the LoS in the GW and ICU increased with age. Finally, the LoS in both GW and ICU beds varied along epidemic waves. It was lower in epidemic valleys than peaks, and decreased after vaccinations began in Colombia.

The data available did not include information in regard inter-hospital transfers. Nevertheless, we were able to follow the whole path of hospital locations of patients reported in the database until their discharge due to recovery or death.

### LoS by bed type and bed pathway

The bed types and bed pathways were similar to those reported in previous studies [6, 8]. Our data also revealed that the majority of patients occupied a GW bed without entering into an ICU, similar to patients in England [6]. This may be due to the low severity of COVID-19 [5]. The majority of the LoS distributions in each bed type and each bed pathway were well fitted to log-normal distributions. As in previous studies, the LoS distributions were right-skewed due to a minority of patients with long hospital stays [6, 8, 10, 18]. The tail of the distribution

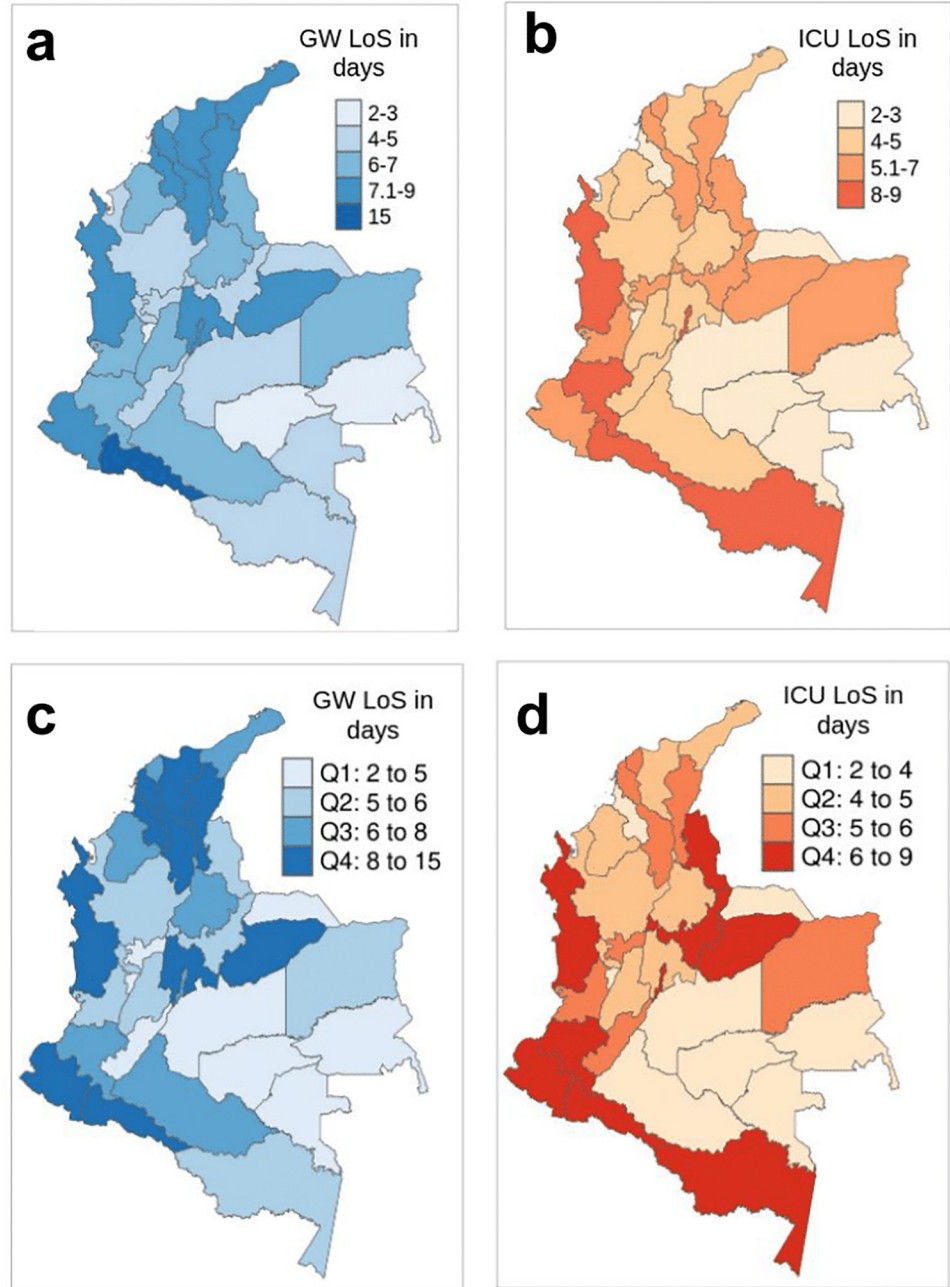

**Fig 2. Patients regions of origin according to similarity in the median of LoS.** (a). Clustering analysis of regions by the median in GW bed (BP1), (b). Clustering analysis of regions by the median in ICU bed (BP2), (c). The median of LoS in GW bed (BP1) for each region classified by quartile, (d). The median of LoS in ICU bed (BP2) for each region classified by quartile.

should not be ignored since these few patients can block beds for a long time and form a heavy burden on capacity [9].

Both the median and IQR of GW and ICU LoS for Colombia patients are similar to the previously reported. However, our IQRs had an upper limit higher than those reported previously. For instance, it was reported that the median (IQR) of the total hospital stay is 14 (10–19) days in China, whereas it was 5 (3–9) days outside of China [9]. A study in New York reported

comparable estimates for total LoS (i.e., median 4.5, IQR 2.4–8.1) [19]. Studies from Northern Italy [19, 20], Japan [21], and California and Washington [22] reported longer estimates of LoS than our data. On the other hand, the median (and IQR) of the ICU stay is 8 (5–13) for China and 7 (4–11) outside of China [9, 23]. In England, the mean ICU length of stay is 12.4 days [8].

## Association between LoS and covariates

We found that there was a similar behaviour for some covariates (i.e., outcome, age, sex, and peaks and valleys) in the model adjusted by all of them and the models adjusted by one at time (or unadjusted model). However, one of the most important differences was the inverted results with respect to the vaccination period. For the first one, the LoS in GW and ICU were lower in the vaccination period than before it, but for the second one they were bigger.

We showed that recovery patients spent more time in GW or ICU beds than deceased patients. This was also found in China [9], and Turkey [5]; different results were found for California and Washington patients with a hospital stay median of 9.3 and 12.17 for survivors and non-survivors, respectively [22]. Also different results were found for London in ICU occupancy [24]. However, some argue that the outcome does not have practical implications to estimate the bed occupancy and influence decision-making [9].

Our data found that older people spent more time in GW and ICU beds than young people, similar results were also reported in previous studies in hospitals in England and Turkey [5, 8]. Also similar to previous studies, men stay longer in GW and ICU beds than women [5, 18]. Some studies concluded that the effect of sex was small and non-significant [8, 25].

According to the model adjusted both the GW and ICU LoS increased during the epidemiological waves. However, these LoS were lower in the vaccination period than before it. Studies in South Africa, Belgium, England, US and Canada also concluded that the LoS in hospitals decrease throughout time [8, 10, 18, 26]. However, none of these studies had an explanation for this tendency. Here, we identified that both the number of admitted patients and the age distribution of patients throughout the waves could possibly affect the LoS (S4 Fig). For instance, the LoS increased/decreased with the increase/decrease in the number of hospital admissions and the increase/decrease in the median age of the admitted patients. We also hypothesized that the decrease in hospital LoS as reported in other studies might be related to improved clinical experience and improved treatments over the course of the epidemic [10, 18, 27].

The GW and ICU LoS were lower in valleys than in peaks. Finally, we found different LoS for patients coming from different geographical regions in Colombia. As previously mentioned, this could be due to different patients profiles (i.e., percentage of comorbidities, ages group, sex), and socioeconomic disparities [28]. It is important to mentioned that the patients region of origin was not always the region in which they get the medical attention. For instance, some regions have very few or none ICU (i.e., Amazonas, Guaviare, Guania, Vaupes and Vichada) [29]. We found that some regions such as Bogota with the highest number of ICU beds in Colombia, is in the highest median LoS quartile in ICU [29]. We also suggest a more deep analysis that shed light on the factors behind this differences observed in our study.

## Contributions

Although LoS is an important factor in predicting the needs of hospital resources, it was rarely a primary outcome in previous studies [18]. Most of the studies estimating LoS were derived from China [9], and they reported the LoS for total hospital and ICU stay without describing the bed pathway [9].

To the best of our knowledge, there are no previous studies of the LoS in Colombia. While there are some similarities with previous studies in the LoS and factors associated with them,

there are also some differences. This discrepancy is due to countries differing in the characteristics of factors associated with LoS [18]. For instance, countries differ in their demographics characteristics, in the management of the pandemic, and have different COVID-19 care guidelines (i.e., the criteria for hospital admission and release might be distinct in different countries) [9, 10].

On the other hand, any prediction model of bed occupancy is sensitive to the value and distribution being assumed for LoS, with strong implications for policy and planning [9]. Therefore, there is a need for local information on LoS for COVID-19 patients [18]. This is important to set with appropriate LoS values the local models that predict the number of beds required and then to provide accurate predictions [18, 30].

We also highlight the usefulness of publicly available data to contribute in answering questions of interest in public health. This prompts to keep building data of open access and improve their record.

## Limitations

This study is based on a secondary analysis of publicly available data that limits the availability of data of what was collected during the pandemic in Colombia. Also, due to the nature of the data, information or selection bias such as missed information or misclassification of the BPs is possible. Finally, there are other factors that may be considered as potential confounders affecting the LoS that were not included in the analysis, such as patient comorbidities, other health conditions or social determinants of health that were not available in the public database.

## Conclusion

While our results share some similarities with previous studies in other countries, the differences in the LoS and the factors affecting the LoS highlight the necessity of local information to parameterize models and improve the predictions of the hospital bed occupancy in Colombia. Future studies might consider comorbidities information of the patients. Finally, these findings help to prepare for risk prevention and control in advance according to the local demographics features and the current situations of medical resources.

## Supporting information

**S1 Fig. Estimation of the median, IQR, and their 95% confidence interval of each AFT model.**
(TIF)

**S2 Fig. Estimation of waves of the epidemic curve of SARS-CoV-2 daily infected cases in Colombia.** (a). The epidemic curve (black) is normalized with respect to the maximum number of cases. The first derivative curve (blue) is also normalized with respect to the maximum and minimum velocity value for positive and negative velocities, respectively. Gray line indicates the zero values. (b). The red line represents the amount of velocity points between the thresholds for each set of points (i.e., each daily velocity and three velocity points before and after each daily point). The velocity thresholds are 150 and -150, those are considered as low velocities of increment and decrement, respectively. The curve is also normalized by the maximum number of the counting. (c). The dashed lines in black indicate the start and the end of an epidemic wave. (d). Epidemic waves (gray bars are time windows not included).
(TIF)

**S3 Fig. Estimation of peaks and valleys of the epidemic curve of SARS-CoV-2 daily infected cases in Colombia.** (a). The epidemic curve (black) is normalized with respect to the maximum number of cases. The first derivative curve (blue) is also normalized with respect to the maximum and minimum velocity value for positive and negative velocities, respectively. Gray line indicates the zero values. (b). The red line (the counting line) represents the amount of velocity points above or below the positive and negative threshold, respectively, for each set of points (i.e., each daily velocity and three velocity points before and after each daily point). The velocity thresholds are 200 and -200, those are considered as high velocities of increment and decrement, respectively. The curve is also normalized by the maximum number of the counting. (c). The dashed lines in black indicate the start and the end of a peak and valley. (d). Epidemic peaks (white) and valleys (pink).
(TIF)

**S4 Fig. Age distribution and admission number throughout the waves.** (a). Age distribution of patients with BP1 (only hospital bed), (b). Age distribution of patients with BP2 (only ICU bed), (c). Admission number of patients with BP1 (only hospital bed), (d). Admission number of patients with BP2 (only ICU bed).
(TIF)

**S1 Table. Acceleration factors of the length of stay in each bed type and each bed pathway.** Those were calculated in different AFT models for bed pathway, outcome, age, sex, waves, peaks and valleys, and vaccination period.
(DOCX)

**S2 Table. Length of stay for GW (BP1) and ICU (BP2) pathways according to covariates.** Those were calculated in different AFT models for bed pathway, outcome, age, sex, waves (W), peaks and valleys, and vaccination period. We get the lognormal distributions parameters from each AFT model and performance sampling to estimate the median and IQR.
(DOCX)

**S3 Table. Median and IQR of the length of stay in GW (BP1) and ICU (BP2).** We get the Lognormal distributions parameters from each AFT model and performance sampling to estimate the median and IQR.
(DOCX)

**S1 Text.**
(DOCX)

**S1 File.**
(DOCX)

## Acknowledgments

We thanks the ReFReCA project which is financed by the Ministry of Science, Technology and Innovation of Colombia—Minciencias (code 111584467754). We also thanks the Grupo de Fundamentos y Enseñanza de la Física y los Sistemas Dinámicos (FEnFisDi) at the Physics Institute of the Universidad de Antioquia for let us use their server to run some simulations.

## Author Contributions

**Conceptualization:** Lina Marcela Ruiz Galvis, Juan Pablo Pérez Bedoya, Oscar Ignacio Mendoza Cardozo.

**Data curation:** Lina Marcela Ruiz Galvis, Carlos Andrés Pérez Aguirre.

**Formal analysis:** Lina Marcela Ruiz Galvis, Carlos Andrés Pérez Aguirre.

**Investigation:** Lina Marcela Ruiz Galvis.

**Methodology:** Lina Marcela Ruiz Galvis, Juan Pablo Pérez Bedoya, Oscar Ignacio Mendoza Cardozo, Juan Pablo Sánchez Escudero, Johnatan Cardona Jiménez.

**Project administration:** Paula Andrea Diaz Valencia.

**Resources:** Lina Marcela Ruiz Galvis, Paula Andrea Diaz Valencia.

**Software:** Lina Marcela Ruiz Galvis, Carlos Andrés Pérez Aguirre.

**Supervision:** Noël Christopher Barengo, Paula Andrea Diaz Valencia.

**Validation:** Lina Marcela Ruiz Galvis, Juan Pablo Sánchez Escudero, Johnatan Cardona Jiménez, Paula Andrea Diaz Valencia.

**Visualization:** Lina Marcela Ruiz Galvis, Noël Christopher Barengo.

**Writing – original draft:** Lina Marcela Ruiz Galvis.

**Writing – review & editing:** Lina Marcela Ruiz Galvis, Noël Christopher Barengo, Paula Andrea Diaz Valencia.

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
