## [Decision Letter · Decision Letter 0]

20 Feb 2023

PONE-D-22-30859Hospital length of stay throughout bed pathways and factors affecting this time: a non-concurrent cohort study of Colombia COVID-19 patients and an unCoVer network project

PLOS ONE

Dear Dr. Ruiz Galvis,

Thank you for submitting your manuscript to PLOS ONE. After careful consideration, we feel that it has merit but does not fully meet PLOS ONE’s publication criteria as it currently stands. Therefore, we invite you to submit a revised version of the manuscript that addresses the points raised during the review process.

This an interesting paper addressing a hot topic. However, some issues can be raised. Regarding study population, please define more clearly inclusion criteria. Were enrolled only patients who resulted positive on admission or also those who became positive during hospital stay? If so, the latter subgroups should be analyzed separately we suggst to better define bed types. Finally we suggest to add more information on interhospital transfer. 

A rebuttal letter that responds to each point raised by the academic editor and reviewer(s). You should upload this letter as a separate file labeled 'Response to Reviewers'.A marked-up copy of your manuscript that highlights changes made to the original version. You should upload this as a separate file labeled 'Revised Manuscript with Track Changes'.An unmarked version of your revised paper without tracked changes. You should upload this as a separate file labeled 'Manuscript'

We look forward to receiving your revised manuscript.

Kind regards,

Chiara Lazzeri

Academic Editor

PLOS ONE

Journal Requirements:

Reviewers' comments:

Reviewer's Responses to Questions

**Comments to the Author**

1. Is the manuscript technically sound, and do the data support the conclusions?

Reviewer #1: Yes

2. Has the statistical analysis been performed appropriately and rigorously? 

Reviewer #1: Yes

3. Have the authors made all data underlying the findings in their manuscript fully available?

Reviewer #1: Yes

4. Is the manuscript presented in an intelligible fashion and written in standard English?

Reviewer #1: Yes

5. Review Comments to the Author

Reviewer #1: In this paper, the authors give a useful description and analysis of the hospital length of stay of patients with COVID-19 in Colombia. Such estimates are essential to understand how disease translates to healthcare resources utilization, and to inform mathematical models which aim to forecast hospital bed occupancy. The authors also discuss some associations between covariates and length of stay, and compare their estimates with previously published ones. Overall, I think this paper is relevant, and I recommend it for publication. I do have some major requests for clarification regarding the limitations of the data and for a change to present the adjusted model in the main manuscript instead of the supplementary material, as well as some minor suggestions. My comments addressed to the authors are listed below, arranged according to their order of appearance in the paper, and with the label [major] to indicate key issues.

[major] General comment: your choice to name the bed types “hospital” and “intensive care unit” introduces some confusion throughout the paper, since these are actually both in hospital. For example, the meaning of phrase “Most of the admitted COVID-19 patients occupied just hospital bed” in the Abstract may not be clear to all, and there may be some confusion between what you refer to as total hospital length of stay versus length of stay in a hospital bed. I would recommend you change the name of “hospital bed” throughout the manuscript, maybe to “general ward bed”, or “regular bed” which you use on line 48.

Line 33: “became shorter after vaccinations began in Colombia than before” -> “decreased after vaccinations began in Colombia” (also applies to line 220).

[major] Line 70: it’s not clear to me if this dataset only includes patients positive for SARS-CoV-2, or also includes patients who were admitted and then subsequently tested positive for SARS-CoV-2. If the latter are included, then did you only look at LoS after the positive test? I’m notably thinking about nosocomial cases here, which may stay in hospital for some time before testing positive. Perhaps a brief mention of this here or in the Discussion would be helpful, as I am not sure to what extent nosocomial transmission of SARS-CoV-2 was a problem in Colombia.

Line 82: I think it would be clearer to introduce the two different bed types that could be occupied by patients here, instead of on line 137. In addition, a brief description of these bed types would be beneficial for an international audience whose healthcare system may be structured differently (e.g. the UK distinguishes between general ward, intensive care unit, and high dependency unit beds).

Line 82: on the topic of bed types – were these the only two categories initially present in the data, or were there more categories that you chose to regroup into these two? (again, here I’m thinking with the UK example in mind, and how high dependency and intensive care units are sometimes grouped together).

[major] Line 84: do you have any information on how inter-hospital transfers may be represented in the data? For example, if a patient is initially admitted in a regular bed in hospital 1, then transferred to an ICU bed in hospital 2, would this be recorded as two separate hospitalisations? If so, and if this occurs frequently, it would severely bias the analysis towards identifying BPs with only one stage and short LoS, so some clarification here/in the Discussion is essential.

Line 134: maybe rephrase to “They represented 5.03% of the cumulative COVID-19 cases in Colombia over that same period”, to remove potential confusion with point prevalence on August 17th specifically.

Table 1: typo in BP1 row, one median is 6.48 and the other is 6.49, these should be the same.

Table 1: N/A would be better defined as “not applicable” in the legend. This is because N/A is for steps which don’t exist in pathways, rather than for data not available to estimate these steps.

Table 2: BP1 is mentioned as the basal group for BP, but if I understand correctly BP1 is basal group for general ward LoS only, and BP2 is basal group for ICU LoS?

[major] Table 2: In your results, there is an important difference between the unadjusted and adjusted models: the association between vaccination period and LoS is reversed. The problem is that you currently present the result of the unadjusted model, briefly mention that the association is reversed in the adjusted model, but then only highlight the result of the adjusted model in your Discussion and Abstract, which is misleading. I would tend to trust more the adjusted model in any case, since several of your covariates are likely associated in some way (e.g. outcome and age). For all of these reasons, I think I think that you should replace Table 2 with something like Table S2, which presents the results for a model adjusted by all covariates. You can then move Table 2 to Supplementary, and comment on this reversed association between vaccination period and LoS as an example of the importance of using an adjusted model.

Line 214: typo “depends”

Line 202: typo “origin” (also line 216)

[major] Line 216: I don’t think you can include geographical region in this sentence, as the analysis is really distinct from the other covariates which you analysed in a joint model. I suggest you instead rephrase this into two sentences. The first to state that LoS depended on outcome, age, sex etc… And the second to state that you see differences in LoS between regions, and that this prompts further investigation into how the covariates mentioned in the first sentence are distributed geographically (see also line 275, which may be a good place to emphasize this).

Line 302: I agree that an important limitation of your analysis here is the data, however it is also a strong point of your work. There is definitely value in highlighting the usefulness of publicly available data for this type of analysis, but this discussion is not currently present in the manuscript.

Figure 1: BP6 is not visible, change box colour to a lighter blue or text to white.

Figure 2: I think the usefulness of this Figure is currently limited. The clustering algorithm does show that there is variation in LoS according to geography, but no other real conclusions can be obtained from this. However, it would be interesting to discuss if the groupings of regions with high/low general ward LoS and high/low ICU LoS are similar. This could be shown more clearly by adding 2 other panels to this Figure, dividing the LoS distribution into quantiles and labelling the regions accordingly.

6. PLOS authors have the option to publish the peer review history of their article (what does this mean?). If published, this will include your full peer review and any attached files.

Reviewer #1: **Yes: **Quentin Leclerc

---

## [Author Response · Author response to Decision Letter 0]

15 Mar 2023

General comment from the editor:

 1. Comment: Regarding study population, please define more clearly inclusion criteria. Were enrolled only patients who resulted positive on admission or also those who became positive during hospital stay? If so, the latter subgroups should be analysed separately 

We have revised that paragraph accordingly. The new paragraph reads now as follows (lines 68-75):

Publicly available data of all registered COVID-19 hospital patients from March 15th 2020 to August 17th 2021 were retrieved from the Colombia National Health Institute [11]. A person was recorded in this database once it was diagnosed with COVID-19. This means, the track of the patients’ bed pathway was made from the moment they became positive for SARS-CoV-2. The diagnosis of SARS-CoV-2 was confirmed in all patients by diagnostic tests [12]. From this database, we only included in-hospital patients diagnosed with COVID-19 who either recovered or deceased. Recovered patients were those with a negative test result and/or with 21 days without symptoms since the symptoms onset [12].

 2. Comment: we suggest to better define bed types. 

According to your suggestion, we have now better defined “bed types” (lines 83-88):

From the database, COVID-19 patients in Colombia were classified as being in-hospital or ICU patients. It means, they could occupy two types of bed: a general ward (GW) bed or an ICU bed, respectively. Unlike a GW bed, an ICU bed implies intensive and specialized medical and nursing care, an enhanced capacity for monitoring, and multiple modalities of physiologic organ support [14].

 3. Comment: Finally we suggest to add more information on interhospital transfer.

According to your suggestion, we have now added more information in regard interhospital transfer (lines 245-248):

The data available did not include information in regard inter-hospital transfers. Nevertheless, we were able to follow the whole path of hospital locations of patients reported in the database until their discharge due to recovery or death.

Review Comments to the Author

Reviewer #1:

 1. [major] General comment: your choice to name the bed types “hospital” and “intensive care unit” introduces some confusion throughout the paper, since these are actually both in hospital. For example, the meaning of phrase “Most of the admitted COVID-19 patients occupied just hospital bed” in the Abstract may not be clear to all, and there may be some confusion between what you refer to as total hospital length of stay versus length of stay in a hospital bed. I would recommend you change the name of “hospital bed” throughout the manuscript, maybe to “general ward bed”, or “regular bed” which you use on line 48.

We change “hospital bed” for “general ward bed” throughout the manuscript. The changes are marked in the text. 

 2. Line 33: “became shorter after vaccinations began in Colombia than before” -> “decreased after vaccinations began in Colombia” (also applies to line 220).

We made this change in line 34 and 243.

 3. [major] Line 70: it’s not clear to me if this dataset only includes patients positive for SARS-CoV-2, or also includes patients who were admitted and then subsequently tested positive for SARS-CoV-2. If the latter are included, then did you only look at LoS after the positive test? I’m notably thinking about nosocomial cases here, which may stay in hospital for some time before testing positive. Perhaps a brief mention of this here or in the Discussion would be helpful, as I am not sure to what extent nosocomial transmission of SARS-CoV-2 was a problem in Colombia.

The data base only includes patients positive for SARS-CoV-2. We do not have information in regard nosocomial transmission during the hospital stay. We have clarified this now in lines 68-75:

Publicly available data of all registered COVID-19 hospital patients from March 15th 2020 to August 17th 2021 were retrieved from the Colombia National Health Institute [11]. A person was recorded in this database once it was diagnosed with COVID-19. This means, the track of the patients’ bed pathway was made from the moment they became positive for SARS-CoV-2. The diagnosis of SARS-CoV-2 was confirmed in all patients by diagnostic tests [12]. From this database, we only included in-hospital patients diagnosed with COVID-19 who either recovered or deceased. Recovered patients were those with a negative test result and/or with 21 days without symptoms since the symptoms onset [12].

 4. Line 82: I think it would be clearer to introduce the two different bed types that could be occupied by patients here, instead of on line 137. In addition, a brief description of these bed types would be beneficial for an international audience whose healthcare system may be structured differently (e.g. the UK distinguishes between general ward, intensive care unit, and high dependency unit beds).

According to your suggestion, we have now better defined “bed types” (lines 83-88):

From the database, COVID-19 patients in Colombia were classified as being in-hospital or ICU patients. It means, they could occupy two types of bed: a general ward (GW) bed or an ICU bed, respectively. Unlike a GW bed, an ICU bed implies intensive and specialized medical and nursing care, an enhanced capacity for monitoring, and multiple modalities of physiologic organ support [14].

 5. Line 82: on the topic of bed types – were these the only two categories initially present in the data, or were there more categories that you chose to regroup into these two? (again, here I’m thinking with the UK example in mind, and how high dependency and intensive care units are sometimes grouped together).

We used the categories available in the data base that only distinguishes between to bed types. in the data. Lines 83-86 read now as follows:

From the database, COVID-19 patients in Colombia were classified as being in-hospital or ICU patients. It means, they could occupy two types of bed: a general ward (GW) bed or an ICU bed, respectively. Unlike a GW bed, an ICU bed implies intensive and specialized medical and nursing care, an enhanced capacity for monitoring, and multiple modalities of physiologic organ support [14].

 6. [major] Line 84: do you have any information on how inter-hospital transfers may be represented in the data? For example, if a patient is initially admitted in a regular bed in hospital 1, then transferred to an ICU bed in hospital 2, would this be recorded as two separate hospitalisations? If so, and if this occurs frequently, it would severely bias the analysis towards identifying BPs with only one stage and short LoS, so some clarification here/in the Discussion is essential.

According to your suggestion, we have now added more information in regard interhospital transfer (lines 245-248):

The data available did not include information in regard inter-hospital transfers. Nevertheless, we were able to follow the whole path of hospital locations of patients reported in the database until their discharge due to recovery or death.

 7. Line 134: maybe rephrase to “They represented 5.03% of the cumulative COVID-19 cases in Colombia over that same period”, to remove potential confusion with point prevalence on August 17th specifically.

We made this change in line 141-142

 8. Table 1: typo in BP1 row, one median is 6.48 and the other is 6.49, these should be the same.

 We made this change in Table 1.

 9. Table 1: N/A would be better defined as “not applicable” in the legend. This is because N/A is for steps which don’t exist in pathways, rather than for data not available to estimate these steps.

We made this change in Table 1.

 10. Table 2: BP1 is mentioned as the basal group for BP, but if I understand correctly BP1 is basal group for general ward LoS only, and BP2 is basal group for ICU LoS?

This is correct, we have now explained this better in the text of S2 Table (previously named Table 2, see the next comment to understand the change in the name):

The basal groups are BP1 (for BPs when comparing the GW LoS), BP2 (for BPs when comparing the ICU LoS).

 11. [major] Table 2: In your results, there is an important difference between the unadjusted and adjusted models: the association between vaccination period and LoS is reversed. The problem is that you currently present the result of the unadjusted model, briefly mention that the association is reversed in the adjusted model, but then only highlight the result of the adjusted model in your Discussion and Abstract, which is misleading. I would tend to trust more the adjusted model in any case, since several of your covariates are likely associated in some way (e.g. outcome and age). For all of these reasons, I think I think that you should replace Table 2 with something like Table S2, which presents the results for a model adjusted by all covariates. You can then move Table 2 to Supplementary, and comment on this reversed association between vaccination period and LoS as an example of the importance of using an adjusted model.

In lines 109-112 we corrected this in the methodology section. The text reads now as follows:: 

We run AFT models adjusted for all covariates for each one of the two most common BPs. We also run an AFT models adjusted for each covariate at time for the four most common BPs, which means that, for each covariate an AFT model was run for each stage of each bed pathway.

In lines 170-176 we now describe the results of the Table 2 (previously named S2 Table):

Table 2 shows that recovered patients spent 2.00 (θ, 95% C.I 1.97-2.03) and 1.79 (θ, 95% C.I 1.74-1.85) more time in GW (BP1) and ICU (BP2) beds, respectively, than deceased patients. The LoS in both beds increased with age, but that increase was non-monotonic. Men spent more time than women in both beds (GW: 1.04 θ, 95% C.I 1.03-1.05 and ICU: 1.07 θ, 95% C.I 1.05-1.10). The LoS increased throughout epidemic waves but decreased in the vaccination period with respect to non-vaccination period. Finally, they were lower in valleys than peaks (GW: 0.58 θ, 95% C.I 0.57-0.59 and ICU: 0.68 θ, 95% C.I 0.66-0.70).

In lines 193-195, 2011-212 we explain the difference between the results in the models adjusted by all covariates and unadjusted:

The results for BP1 and BP2 with the model adjusted by all covariates and adjusted by one covariate at time were similar in the behaviour of covariates except for the waves and the vaccination period. 

And

Contrary to the model adjusted by all covariates, for a model just adjusted by vaccination period we found that: ….

We re-wrote lines 271-277 to highlight the results of the model adjusted for all covariables as the main result:

We found that there was a similar behaviour for some covariates (i.e., outcome, age, sex, and peaks and valleys) in the model adjusted by all of them and the models adjusted by one at time (or unadjusted model). However, one of the most important differences was the inverted results with respect to the vaccination period. For the first one, the LoS in GW and ICU were lower in the vaccination period than before it, but for the second one they were bigger. 

We also modified the Supplementary Information to change the S1 table (previously named Table 2) and S2 Table (previously named S1 Table).

 12. Line 214: typo “depends”

 We made this change in line 221

 13. Line 202: typo “origin” (also line 216)

We made this change in line 209 y 223

 14. [major] Line 216: I don’t think you can include geographical region in this sentence, as the analysis is really distinct from the other covariates which you analysed in a joint model. I suggest you instead rephrase this into two sentences. The first to state that LoS depended on outcome, age, sex etc… And the second to state that you see differences in LoS between regions, and that this prompts further investigation into how the covariates mentioned in the first sentence are distributed geographically (see also line 275, which may be a good place to emphasize this).

In lines 237-239 we create new sentences to state the difference in LoS between regions: 

There were also differences in LoS throughout the geographical regions from which the patients origin, which prompts further investigation into how the covariates mentioned previously are distributed geographically

 15. Line 302: I agree that an important limitation of your analysis here is the data, however it is also a strong point of your work. There is definitely value in highlighting the usefulness of publicly available data for this type of analysis, but this discussion is not currently present in the manuscript.

 We have now highlighted this in lines 331 and 333 of the manuscript: 

We also highlight the usefulness of publicly available data to contribute in answering questions of interest in public health. This prompts to keep building data of open access and improve their record.

 16. Figure 1: BP6 is not visible, change box colour to a lighter blue or text to white.

This change was made in the figure 1.

 17. Figure 2: I think the usefulness of this Figure is currently limited. The clustering algorithm does show that there is variation in LoS according to geography, but no other real conclusions can be obtained from this. However, it would be interesting to discuss if the groupings of regions with high/low general ward LoS and high/low ICU LoS are similar. This could be shown more clearly by adding 2 other panels to this Figure, dividing the LoS distribution into quantiles and labelling the regions accordingly.

We have now added to the methodology this additional information (lines 131-132): 

We also divided the distribution of the median LoS of geographical regions into quartiles and labelled the regions accordingly.

Changes in lines 220-225 of the results section (and also modified in Fig 2): 

Figs 2c and 2d present the regions classified by the quartiles of the median distribution of LoS. They clearly show the regions with both high GW and ICU LoS (i.e., Nariño, Putumayo, Chocó, and Casanare), and both low GW and ICU LoS (i.e., Meta, Guaviare, Vaupés, Guainía, Arauca). Also, people from certain regions had median LoS in the lowest/highest quartile in GW beds but the highest/lowest quartile in ICU beds (i.e., Sucre).

And finally, we re-wrote lines 303-311:

Finally, we found different LoS for patients coming from different geographical regions in Colombia. As previously mentioned, this could be due to different patients’ profiles (i.e., percentage of comorbidities, ages group, sex), and socioeconomic disparities [28]. It is important to mentioned that the patient’s region of origin was not always the region in which they get the medical attention. For instance, some regions have very few or none ICU (i.e., Amazonas, Guaviare, Guania, Vaupes and Vichada) [29]. We found that some regions such as Bogota with the highest number of ICU beds in Colombia, is in the highest median LoS quartile in ICU [29]. We also suggest a more deep analysis that shed light on the factors behind the differences observed in our study.

 Additional changes

 1. We change the spelling of the name of the last author.

 2. We change the footnote of Table 1 and 2. 

 3. We incorporate reference 14 and 29 according to the reviewer’s suggestions.

---

## [Editor Report · Decision Letter 1]

20 Mar 2023

Hospital length of stay throughout bed pathways and factors affecting this time: a non-concurrent cohort study of Colombia COVID-19 patients and an unCoVer network project

PONE-D-22-30859R1

Dear Dr. Ruiz Galvis,

We’re pleased to inform you that your manuscript has been judged scientifically suitable for publication and will be formally accepted for publication once it meets all outstanding technical requirements.

Kind regards,

Chiara Lazzeri

Academic Editor

PLOS ONE
---

## [Editor Report · Acceptance letter]

28 Mar 2023

PONE-D-22-30859R1 

Hospital length of stay throughout bed pathways and factors affecting this time: a non-concurrent cohort study of Colombia COVID-19 patients and an unCoVer network project 

Dear Dr. Ruiz Galvis:

I'm pleased to inform you that your manuscript has been deemed suitable for publication in PLOS ONE. Congratulations! Your manuscript is now with our production department. 

Kind regards, 

on behalf of

Dr. Chiara Lazzeri 

Academic Editor

PLOS ONE